# Association of Nitric Oxide Synthase Polymorphism and Coagulopathy in Patients with Osteonecrosis of the Femoral Head

**DOI:** 10.3390/jcm11174963

**Published:** 2022-08-24

**Authors:** Cheng-Ta Wu, Rio L. C. Lin, Pei-Hsun Sung, Feng-Chih Kuo, Hon-Kan Yip, Mel S. Lee

**Affiliations:** 1Department of Orthopaedic Surgery, Kaohsiung Chang Gung Memorial Hospital, Chang Gung University College of Medicine, Kaohsiung 83301, Taiwan; 2Department of Medicine, Division of Cardiology, Kaohsiung Chang Gung Memorial Hospital, Chang Gung University College of Medicine, Kaohsiung 83301, Taiwan; 3Department of Othopaedic Surgery, Pao-Chien Hospital, Pingyung 90064, Taiwan

**Keywords:** osteonecrosis of the femoral head, coagulopathy, nitric oxide synthase, nitric oxide, polymorphism

## Abstract

Genetic polymorphism of nitric oxide synthase (NOS) can cause reduction of nitric oxide (NO) levels and may be associated with osteonecrosis of the femoral head (ONFH). However, the association of coagulopathy and NOS polymorphism in ONFH patients has not been confirmed. Between November 2005 and October 2013, 155 patients with ONFH were recruited in the study of serum coagulation profiles and NOS polymorphism. Another 43 patients who had dysplasia, osteoarthritis, or trauma of hip joints were included as controls. PCR genotyping for the analysis of NOS 27-bp polymorphism in intron 4 was performed. The analysis of coagulation profiles included fibrinogen, fibrinogen degradation product (FDP), protein S, protein C, and anti-thrombin III. The results showed that 27-bp repeat polymorphism was significantly associated with ONFH (OR 4.32). ONFH patients had significantly higher fibrinogen, FDP, protein S, and anti-thrombin III levels than that of the controls. The incidence of coagulopathy was significantly higher in ONFH patients (73.2%), and the odds ratio increased from 2.38 to 7.33 when they had 27-bp repeat polymorphism. Patients with hyperfibrinogenemia, elevated FDP levels, and with the risk factor of alcohol or steroid use had significantly higher risks of bilateral hip involvement. This study demonstrated the presence of NOS polymorphism, and a resultant reduction in NO production was associated with coagulopathy, which in turn might contribute to higher risks of bilateral ONFH. Our data suggests that checking NOS polymorphism and coagulopathy may provide a new avenue in managing ONFH.

## 1. Introduction

Osteonecrosis of the femoral head (ONFH) has a high prevalence of bilateral hip involvement and is associated with various risk factors [1,2,3]. Many reports suggest that thrombophilia or hypofibrinolysis may participate in the pathogenesis of ONFH and result in multifocal osteonecrosis [4,5,6,7,8]. Jones et al. studied 45 ONFH patients and found 82.2% had coagulation abnormality [9]. Zalavras et al. found 22.2% of the 72 ONFH patients had either factor V Leiden or prothrombin mutation, while it was only 7.3% in the 300 control subjects [10]. On the contrary, Lee et al. reported coagulopathy did not exist in Korean patients based on the results in 24 ONFH patients and 24 matched controls [11]. Furthermore, neither factor V Leiden nor prothrombin mutation was found in ONFH patients as well as in the control subjects in an Asian population [12,13,14,15]. These conflicting results are perplexing since the prevalence of ONFH in the Asian population is reported to be higher than in the Caucasian population. Nevertheless, single nucleotide polymorphism in other loci of factor V (ex. rs6020) with concomitant coagulopathy in 87.6% of the ONFH patients were reported in Chinese patients [14,15].

Nitric oxide (NO), a multifunctional free radical that coordinates diverse biologic processes, may possibly participate in the pathogenesis of osteonecrosis [16,17,18,19]. Polymorphism of the NO synthase (NOS) gene with 27-bp repeat polymorphism or with T-786C polymorphism had been found in both Asian and Caucasian patients who had ONFH [17,18,19]. Evidence also showed that NOS polymorphism may cause reduction of NO production, which in turn enhances platelet aggregation, impairs angiogenesis, reduces mobilization of endothelial progenitor cells, and inhibits bone formation and bone volume in mice [20,21,22,23]. Theoretically, these NOS-mediated responses had been postulated to augment the sequence of thrombophilia and hypofibrionolysis [4,5,6,7,17,18,19]. However the association between coagulopathy and NOS polymorphism in patients with ONFH has not been confirmed. Furthermore, it remains unknown if patients with coagulopathy have higher risks for bilateral hip involvement. 

The purpose of this study was to analyze the association between the 27-bp repeat NOS polymorphism, bilateral hip involvement, and the presence of coagulopathy in patients with ONFH in order to better understand the pathogenesis of ONFH.

## 2. Materials and Methods

Between November 2005 and October 2013, 155 patients (mean age 44.5 ± 11 years, 103 male and 52 female) with ONFH were included in the study of serum coagulation profiles and NOS polymorphism. The diagnosis of ONFH was confirmed by radiographs and MRI in all patients. There were 112 patients presented with bilateral lesions. The diseases were staged according to the system of the Association Research Circulation Osseous (ARCO) [24]. There were 29 hips with stage I, 95 hips with stage II, 82 hips with stage III, and 61 hips with stage IV disease. Because healthy subjects may have relatively normal coagulation profiles, we recruited 43 patients (mean age 56.2 ± 12.2 years; 30 male and 13 female) who had dysplasia [17], osteoarthritis [23], or trauma [3] of the hip joints as the control group for comparison. The study had followed the protocols approved by the Institutional Research Committee (IRB94-1024B; IRB97-2214A3; IRB102-2207C; IRB103-6952A3; IRB106-1065C; 202001545B0). Patient demographics, risk factors, and laboratory tests of coagulation profiles were recorded. All patients were examined with standard pelvis radiographs for both hip joints until the latest follow-up. The analysis of coagulation profiles included fibrinogen, fibrinogen degradation product (FDP), protein S, protein C, and anti-thrombin III. An abnormal coagulation profile was defined as any values out of the normal ranges, including fibrinogen < 190 mg/mL or > 380 mg/mL; FDP > 10 μg/mL; proteins S < 90% or > 130%; protein C < 70% or > 140%; anti-thrombin III < 75% or >125% [8,14,15]. There was no familiar relationship between any participants in the study and control group.

For the analysis of NOS 27-bp polymorphism in intron 4, genomic DNA was prepared by using the QIAamp Blood Midi Kit and Maxi Kit (Qiagen, Valencia, CA, USA) from the whole blood of the patients. PCR genotyping for the 27-bp repeat SNP was performed using the primer set (forward 5′-AGGCCCTATGGTAGTGCCTT-3′; reverse 5′-TCTCTTAGTGCTGTGGTCAC-3′). The PCR products were separated by 3% agarose gel electrophoresis. The 420 bp wild-type product contained five 27-bp repeats (the b allele), and the 393 bp mutant type contained four 27-bp repeats (the a allele).

In comparisons of the coagulation profiles, the allele and genotype frequencies between the ONFH patients and the controls were performed using the chi-square test. The results were considered significant when *p* < 0.05. All statistical analyses were performed using SPSS for Windows v11.5.0 (SPSS Inc., Chicago, IL, USA).

## 3. Results

When compared with the controls, the frequency of 4a allele was significantly higher in ONFH patients (13.6% vs. 3.5%, *p* = 0.006, OR 4.37, 95% CI 1.32–14.51). The 4a/4a genotype was found in 2.8% of the ONFH patients but not in the controls. (Table 1). In addition, the ONFH patients showed a higher frequency of the 4a/4a genotype or the 4a/4b genotype than the controls (21.7% vs. 7%, *p* = 0.016, OR 4.32, 95% CI 1.26–14.84). There were no significant changes in allele and genotype frequency between the ONFH patients and the control group regarding gender, unilateral and bilateral AVN, or age (<50 or >50 years old).

Comparisons of the coagulation profiles between the ONFH patients and the controls are listed in the Table 2. The serum fibrinogen levels were significantly higher in the ONFH patients as compared with the controls (341.5 mg/mL vs. 277.4 mg/mL, *p* < 0.001). A significant portion of the ONFH patients were found to have hyperfibrinogenemia (27.5% vs. 2.3%; OR 15.94, 95% CI 2.13–119.65; *p* < 0.001) and elevated FDP levels (47.7% vs. 30%; OR 2.12, 95% CI 1.0–4.49; *p* = 0.05) as compared with the controls. Interestingly, ONFH patients also had higher protein S and anti-thrombin III levels as compared with the controls (*p* < 0.001). In addition, there were significantly more patients in the ONFH group with elevated protein S levels (>130%) than in the control group (15.6% vs. 2.4%; OR 7.61, 95% CI 1.0–58.08; *p* = 0.019). The proportions of patients with protein S deficiency, protein C deficiency, or anti-thrombin III deficiency in the ONFH group were not different from that of patients in the control group. In summary, there was a significantly higher proportion of ONFH patients (73.2%) than controls (OR 2.38, 95% CI 1.18–4.77, *p* = 0.016) having coagulopathy. ONFH patients who had 27-bp repeat polymorphism of the NOS gene had a significantly higher risk of coagulopathy or an abnormal coagulation profile as compared with the controls (OR 7.33 and 9.0, respectively).

We further analyzed the association of coagulopathy, risk factors, or NOS polymorphisms with the incidence of bilateral hip involvement in ONFH patients. (Table 3). We found that patients with hyperfibrinogenemia (elevated fibrinogen levels and FDP levels) and with risk factors of alcohol or steroid use had significantly higher risks of having bilateral ONFH. However, the presence of 4a/4a or 4a/4b genotypes did not correlate with higher risks of bilateral hip involvement.

## 4. Discussion

Thrombophilia or hypofibrinolysis may cause intravascular coagulopathy, deter the blood flow, and result in intraosseous hypoxia and ischemia [5,7]. Jones Jr. and Glueck et al. were the pioneers to discover the potential links between ONFH and coagulopathy [4,25]. Jones LC et al. further reported 82.2% of their patient cohort was identified with coagulation abnormalities [9]. However, Lee et al. argued that coagulopathy did not exist in East Asian patients because no differences were found in the thrombotic factors (protein C, protein S, antithrombin III, anticardiolipin antibody, and lupus antibody) and the fibrinolytic factors (tissue plasminogen activator, plasminogen activator inhibitor-1, lipoprotein (a), and plasminogen) in 24 ONFH patients and 24 matched controls [11]. In the current study, we recruited 155 ONFH patients and found 73.2% of them had abnormal coagulation profiles. This percentage was significantly higher in the ONFH group than in the control group (odds ratio 2.38, *p* = 0.016). The finding was in agreement with other studies, showing that 74% to 87.6% of their patients had coagulation disorders [7,14]. We found the average serum fibrinogen levels in ONFH patients were significantly higher than in the controlled patients (341.5 mg/mL vs. 277.4 mg/mL, *p* < 0.001). There were more ONFH patients presenting with hyperfibrinogenemia (27.5% vs. 2.3%; OR 15.94; *p* < 0.001) and elevated FDP level (47.7% vs. 30%; OR 2.12; *p* = 0.05) as compared with the controls. Notably, protein S or protein C deficiency as well as low anti-thrombin III was not evident in our patients. It is well known that familial protein S deficiency could lead to thrombophilia and venous thrombosis [26]. A low level (<65%) of protein S was found in 23–36% of ONFH patients vs. 2–3% of control subjects [6,18]. However, the current study had contradictory results that ONFH patients were found to have significantly higher protein S and anti-thrombin III levels as compared with the controls. In addition, there were more ONFH patients having elevated protein S levels (>130%), when compared with the controls (15.6% vs. 2.4%; OR 7.61, 95% CI 1.0–58.08; *p* = 0.019). In a large population-based study, patients with venous thrombosis did not have lower protein S levels as compared with those in the age- and sex-matched healthy controls [27]. The Second Northwick Park Heart Study, with a 14-year follow-up on 3052 middle-aged men, also confirmed that patients with higher protein S levels were associated with future risks of coronary heart disease [28]. Using the National Health Insurance Research Database, Sung et al. found ONFH patients had significantly higher risks for cardiovascular, cerebrovascular, or venous thromboembolic events [29,30]. Although the underlying mechanism of the higher protein S levels in ONFH is not clear, it is possible that the elevated protein S and anti-thrombin III levels may be secondary to or associated with endothelial dysfunction and intraosseous thromboembolism.

NO is a versatile molecule produced by the enzymatic action of NOS, and it has been identified to participate in the pathogenesis of ONFH [16]. Single nucleotide polymorphisms of NOS, particularly 27-bp repeat polymorphism and T-786C polymorphism, can result in the reduction of NO production, and it is associated with higher risks of ONFH [17,18,19]. We found the frequency of 4a allele was significantly higher in ONFH patients than in the controls (13.6% vs. 3.5%, *p* = 0.006, OR 4.37, 95% CI 1.32–132 14.51). The frequency of 4a/4a and 4a/4b genotypes was also significantly higher in ONFH patients than in the controls (24.5% vs. 7%, *p* = 0.016, OR 4.32, 95% CI 1.26–14.84). Reduced NO production resulted in reduced vascular reactivity, inhibited angiogenesis, impaired recruitment of endothelial progenitor cells, and lower bone mass in clinical subjects and animal experiments [16,17,18,19,21,22,23]. NO has antithrombotic activity by limiting platelet activation, adhesion, aggregation, and recruitment to the thrombus [21,22]. Theoretically, coagulopathy could be detected in ONFH patients with NOS polymorphism. However, previous studies did not confirm this association. This study found ONFH patients with 27-bp repeat polymorphism had a significantly higher risk of coagulopathy (OR 7.33) or an abnormal coagulation profile (OR 9.0) as compared with the controls. The microstellate polymorphism of 4a allele and the resultant reduction of NO production may be associated with or contribute to the coagulopathy and the pathogenesis of ONFH. 

Glueck et al. reported coagulopathy in 31–61% of ONFH patients, and 49% of them had bilateral hip involvement [5]. Jones et al. found 82.2% of ONFH patients had coagulopathy and 83% had bilateral hips involved (34 of the 41 hips) [9]. In an exploratory study on 26 patients with multifocal osteonecrosis matched with 91 unifocal osteonecrosis and 117 normal controls, thrombophila and NOS T-786C polymorphism were found to be associated with multifocal osteonecrosis. However, no differences were found between multifocal and unifocal osteonecrosis [19]. Similarly, this study found the 27-bp repeat polymorphism was not associated with a higher incidence of bilateral ONFH. Nevertheless, patients with hyperfibrinogenemia (elevated fibrinogen levels and FDP levels) and with the risk factor of alcohol or steroid use had significantly higher risks of having bilateral ONFH.

In this study, we did not recruit healthy subjects as controls because they might have relatively normal coagulation profiles. Instead, we used patients with hip disorders as controls and found 53.5% of them had coagulopathy. The presence of coagulation abnormalities in the control subjects is not surprising. Jones et al. used healthy individuals as the control group and found 30% of them had 1 or 2 abnormal serologic results for coagulation profile tests [9]. Glueck et al. found that 38% of healthy women with prior pregnancies had tested positive for coagulation abnormalities [31]. These implied that some individuals might have a coagulation abnormality trait that can precipitate thrombotic or hypofibrinolytic events after exposure to an inducing factor. It may also be possible that disorder in the hip joint may release tissue factors to trigger an extrinsic coagulation pathway and result in changes of the serum coagulation profile. 

There were some limitations in this study. First, the study was not a population-based genome-wide association study with a large enough sample size. We recruited only 155 ONFH patients and 43 controls. Second, the controls were not sex- and age-matched. Third, all subjects belonged to one ethnic group. Nevertheless, our results were congruent with previous studies using healthy or matched control subjects in different ethnic groups [7,9,12,13,14,15,16,17,18,19]. The frequency of the 4a allele in our controls was 3.5%, and it was also similar to the frequency (2.4%) in 206 healthy subjects reported by Koo et al [17]. The only difference was the lack of protein S and anti-thrombin III deficiency in our study.

## 5. Conclusions

We confirmed that 27-bp repeat microstellate polymorphism of NOS was associated with ONFH and coagulopathy in Taiwanese patients. ONFH patients with coagulopathy and with the risk factor of alcohol or steroid use were associated with higher risks of bilateral hip involvement. This study suggested that NOS polymorphism with low NO levels may contribute to the coagulopathy and the pathogenesis of ONFH.

## Figures and Tables

**Table 1 jcm-11-04963-t001:** NOS 27-bp repeat polymorphisms.

	ONFH ^a^	Control	OR (95% CI) ^b^	*p*-Value
Allele Frequency				
4a	39 (13.6%)	3 (3.5%)	4.37 (1.32–14.51)	0.006
4b	247 (86.4%)	83 (96.5%)		
Genotype Frequency				
4a/4a	4 (2.8%)	0 (0%)	4.32 (1.26–14.84)	0.016
4a/4b	31 (21.7%)	3 (7%)		
4b/4b	108 (75.5%)	40 (93%)		

^a^ Osteonecrosis of femoral head. ^b^ Odds Ratio (95% Confidence Interval).

**Table 2 jcm-11-04963-t002:** Association of Coagulopathy with NOS ^a^ polymorphism and ONFH.

	ONFH ^b^	Control	OR (95% CI) ^c^	*p*-Value
Fibrinogen (mg/mL)	341.5 ± 90.5	277.4 ± 52.7		<0.001
>380 mg/mL	41 (27.5%)	1 (2.3%)	15.94 (2.13–119.65)	<0.001
≦380 mg/mL	108 (72.5%)	42 (97.7%)		
FDP elevation ^d^			2.12 (1.0–4.49)	0.05
Yes	71 (47.7%)	12 (30%)		
No	78 (52.3%)	28 (70%)		
Protein S (%)	111.8 ± 22.1	96.8±17.4		<0.001
<90	27 (18.4%)	9 (21.4%)		0.659
>130	23 (15.6%)	1 (2.4%)	7.61 (1.0–58.08)	0.019
Protein C (%)	120 ± 32.4	126.7 ± 59.9		0.339
<70	5 (3.5%)	1 (2.4%)		1.000
>140	32 (22.4%)	7 (16.7%)		0.522
Anti-thrombin III (%)	98.7 ± 20.8	83.5 ± 16		0.000
<75	11 (29.7%)	23 (20%)		0.258
>125	9 (7.8%)	0 (0%)		0.114
Coagulopathy			2.38 (1.18–4.77)	0.016
With	112 (73.2%)	23 (53.5%)		
Without	41 (26.8%)	20 (46.5%)		
Abnormal coagulation profile				0.111
Present	119 (77.8%)	28 (65.1%)		
Absent	34 (22.2%)	15 (34.9%)		
Coagulopathy and 27-bp polymorphism			7.33 (0.94–57.11)	0.026
4a/4a or 4a/4b	26 (25%)	1 (4.3%)		
4b/4b	78 (75%)	22 (95.7%)		
Abnormal coagulation profile and 27-bp polymorphism			9.00 (1.17–69.42)	0.009
4a/4a or 4a/4b	27 (25%)	1 (3.6%)		
4b/4b	81 (75%)	27 (96.4%)		

^a^ Nitric oxide synthase. ^b^ Osteonecrosis of femoral head. ^c^ Odds Ratio (95% Confidence Interval). ^d^ fibrinogen degradation product.

**Table 3 jcm-11-04963-t003:** Risks for Bilateral Hip Involvement in ONFH.^a^

	Unilateral	Bilateral	OR (95% CI) ^b^	*p*-Value
Fibrinogen > 380 mg/mL			3.6 (1.3–9.96)	0.013
Yes	5 (12.2%)	36 (33.3%)		
No	36 (87.8%)	72 (66.7%)		
FDP elevation ^c^			3.41 (1.55–7.5)	0.002
Yes	11 (26.8%)	60 (55.6%)		
No	30 (73.2%)	48 (44.4%)		
Risk factor				0.001
Alcohol	25 (58.1%)	78 (69.6%)		
Steroid	1 (2.3%)	17 (15.2%)		
Idiopathic	17 (39.5%)	17 (15.2%)		
NOS Polymorphisms				0.585
4a/4a or 4a/4b	10 (24.4%)	25 (24.5%)		
4b/4b	31 (75.6%)	77 (75.5%)		

^a^ Osteonecrosis of femoral head. ^b^ Odds Ratio (95% Confidence Interval). ^c^ fibrinogen degradation product.

## Data Availability

Data can be obtained from the corresponding author upon reasonable request.

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
