# Peer review of "Association of Nitric Oxide Synthase Polymorphism and Coagulopathy in Patients with Osteonecrosis of the Femoral Head"

_jcm, 2022, doi:10.3390/jcm11174963_

Round 1

Reviewer 1 Report

Dear Authors,

Thank You for manuscript, I read it with interest and appreciation. Here are some suggestions in order to future improve it and clarify some parts.

1.     Authors should present the methods used to diagnose AVN: X-rays, MRI?

2.     Did authors used any AVN classification: ARCO, Arlet& Ficat? Authors should present the results of one of these classifications. 

3.     There is not any information about the gender in study and control group. 

4.     How many patients from the control group had bilateral and unilateral AVN?

5.     Were there any changes of allele and fenotype frequency in comparison to: gender, unilateral and bilateral AVN, younger (<50 years old) and older (>50 years old) patients?

6.     In my opinion the study group is too small and should contain around 100 healthy participants without any orthopedic diseases. Authors should make a statement that there was any familiar relationship between any participant from study and control group. 

In conclusion, I think your paper is valid but can be further improved after the revision of some aspects.

Reviewer 2 Report

In this the authors have carried out the PCR genotyping for the analysis of NOS 27-bp polymorphism in intron 4.  Patients with dysplasia, osteoarthritis, or trauma of hip joints served as controls.  The authors have also evaluated the parameters like fibrinogen, FDP, Protein S, Protein C and Anti-thrombin III.  Using statistical tests, they have shown that NOS polymorphism is associated with ONFH.  Though many previous studies have shown an association of NOS polymorphism with ONFH an analysis of factors that are associated with coagulopathy has not been performed.

However, the authors need to address the following questions before the paper is accepted.

1.      The normal range of fibrinogen range is 200-400 mg/ml.  The fibrinogen range provided is well within the range though it is still significantly higher than that of controls.

2.      The normal range of other parameters like fibrinogen, FDP, Protein S, Protein C and Anti-thrombin III may please be provided.  A graph for control and ONFH with frequency distribution with normal range might help to better represent the data.  If most of the parameters are with in normal range, then how does the author explain the functional relevance of NOS with coagulation in ONFH.

3.      Since the fibrinogen levels are within the normal range, has the authors checked if the clotting time is significantly different between the two groups.  The data may be provided if it is available.  Though higher fibrinogen levels might be suggestive of this.  The data also have to be compared with healthy controls.

4.      Can the authors provide the fibrinogen, FDP, Protein S, Protein C and Anti-thrombin III levels and the NOS genotype for the patients who are at risk with alcoholic and steroid category.  How does it compare with group of patients who are not in this risk category?

5.      It would have been ideal to include healthy controls with genotyping.  This will also help to evaluate the coagulation parameters and if there are differences.

Round 2

Reviewer 1 Report

The Authors made all proposed changes in the manuscript. My recommendation is to accept this paper in present form.